



# Using Doppler lidar systems to detect atmospheric turbulence in Iceland

Shu Yang[1,2], Guðrún Nína Petersen[2], Sibylle von Löwis[2], Jana Preißler[3], David Christian Finger[1]

1) Reykjavik University, School of Science and Engineering, Reykjavik, Iceland

2) Icelandic Meteorological Office, Reykjavik, Iceland

3) Centre for Climate and Air Pollution Studies, National University of Ireland, Galway, University Road, H91CF50, Galway, Ireland

## Abstract

The temporal and spatial scale of atmospheric turbulence is very dynamic, requiring an adequate method to detect and monitor turbulence with high resolution. Doppler Light Detection and Ranging (lidar) systems have been used widely to observe and monitor wind velocity and atmospheric turbulence profiles. Lidar systems can provide continuous information about wind fields using the Doppler effect from emitted light signals. In this study, we use a Leosphere Windcube 200S lidar system stationed in Reykjavik, Iceland, to evaluate turbulence intensity by estimating eddy dissipation rate (EDR). For this purpose, we retrieved radial wind velocity observations from velocity azimuth display (VAD) scans to compute EDR based on the Kolmogorov theory. We compared different noise filter methods, scan strategies and calculation approaches during different selected weather conditions to assess the accuracy of our EDR estimations. The results reveal that the lidar observations can detect and quantify atmospheric turbulence with high spatial and temporal resolution, our algorithm can retrieve EDR and indicate the turbulence intensity. These results suggest that lidar observation can be of high importance for potential end-user, e.g. air traffic controllers at the local airport. The work is an important step towards enhanced aviation safety in a subpolar climate characterized by severe wind turbulence.

## 1 Introduction

Extreme weather phenomena can have hazardous impacts on aviation safety (Sharman, 2016). In particular, turbulent headwinds during aircraft take-off or landing can lead to critical situations. The headwind is the relative air motion in an opposite sense to the direction of the aircraft, provides the lift of an aircraft(Hon and Chan, 2014). According to the safety report of the International Civil Aviation Authorities (ICAO) , more than one-third of aircraft accidents



in 2017 occurred during take-off and landing (ICAO, 2018). During take-off and landing phases, rapid and large changes in headwind can be a severe threat to aircraft safety. In such situations, the air speed is relatively low and proximity to the surface leaves little room for taking corrective measures (Hon and Chan, 2014).

Atmospheric turbulence is the main cause for this kind of rapid wind change. From topography to weather conditions, several factors may enhance the generation of atmospheric turbulence. Detection of turbulence is a common issue in the aviation sector, here we focus on the detection in Iceland.

Iceland, located in the North Atlantic Ocean, is well known for particular turbulent weather situations, leading frequently to aviation safety concerns. Strong winds and large wind speed variations are frequent in Iceland (Ólafsson and Ágústsson, 2007), and detecting turbulence to enhance aviation safety is notably relevant in Iceland.

By definition, the scale of atmospheric turbulence varies from the macroscale to microscale. The eddies that affect aviation the most have a spatial scale between about 100 m and 1 km (Sharman, 2016). It is hard to detect and monitor turbulence continuously with conventional instruments, such as meteorological masts, which only measure the wind at some certain altitude; or weather balloon, which only measures at some certain time. In order to detect and quantify turbulence and get a clearer picture of the wind conditions, several international airports, e.g. Hong Kong (Chan, 2010), Sendai (Misaka et al., 2008), have installed light detection and ranging (lidar) systems.

The use of lidars for remote sensing of wind has been increasing in the last decade. In addition to being used in the aviation sector (Hon and Chan, 2014; Leung et al., 2018), they are widely used in the wind energy sector (Wächter and Rettenmeier, 2009) as well as in meteorology research (Manninen et al., 2018; Tuononen et al., 2017). Compared to the use of conventional methods like anemometers and radio sounding, lidars have the advantage of continuous measurement at high temporal and spatial resolution. Accordingly, lidar systems allow the monitoring of wind fields within and above the boundary layer without the need of masts, which is especially important for mobile installation. Also, tall masts may not be desirable in some places, e.g. at airports. The lidars can also be mobile and can therefore easily be deployed at different locations, allowing for flexibility for measurements during e.g. an event or campaign.

There is existing research on turbulence detection using Doppler lidar, from theoretical approach to practical research: (Frehlich, 2001; Frehlich et al., 2006; Frehlich and Cornman, 2002) developed a method to estimate turbulence intensity from Kolmogorov theory; In Europe, some studies have been done on retrieving turbulence intensity from lidar datasets (e.g. (O'Connor et al., 2010; Thobois et al., 2015)); At Hong Kong International Airport, lidars have been applied to detect low-level turbulence (Chan, 2009; Hon and Chan, 2014; Leung et al., 2018). However, the use of lidars for turbulence intensity detection in high latitude regions, such as Iceland, has received little attention. An algorithm to retrieve turbulence intensity from the vertical lidar scans, hereafter called vertical stares as the beam is kept in vertical position, has been developed in Finland (O'Connor et al., 2010). Horizontal wind velocity is generally an order of magnitude larger than vertical velocity, and aircrafts with high air speed and wind loading are sensitive to head/tail wind variations (Sinclair and Kuhn, 1991). Accordingly, it is valuable and meaningful to develop an algorithm to retrieve the turbulence intensity from the lidar dataset of horizontal wind velocity in Iceland.

In this study, we derive turbulence intensity by computing the eddy dissipation rate (EDR) from wind profile data, acquired using a Leosphere Windcube 200S Doppler lidar system



(Leosphere, Inc, 2013) located at the Icelandic Meteorological Office (IMO) in Reykjavik,
Iceland. The results are compared to vertical stares using O'Connor et al. (2010)'s method.
And we also compared two calculation approaches and different data filtering methods. The
next section contains information about the lidar system and the dataset. The algorithm to
retrieve turbulence intensity from lidar dataset is described as well. The results are presented
in Sect. 3 and final conclusion and discussions in the last section.
## 2  Methodology
### 2.1  Instruments
Two identical lidar systems are currently in operation in Iceland, Leosphere Windcube 200S
Doppler scanning lidars with a depolarization module. That module can distinguish the shape
of particles in the atmosphere, but will not be applied in this study. One is located at Keflavik
International Airport and the other system is a mobile one installed on a trailer. The mobile
system is currently located at IMO's headquarters in Reykjavik. Here we use data from the
latter system.
Table 1. The specifications of the lidars operated in Iceland (Leosphere, Inc, 2013).

|                          | Specification           |
|--------------------------|-------------------------|
| Company                  | LEOSPHERE GROUP         |
| Website:                 | http://www.leosphere.com |
| Model                    | Windcube 200S           |
| Wavelength               | 1.54 µm                 |
| Maximum Power            | 5 mW                    |
| Pulse Width              | 200 ns                  |
| Range resolution         | 50 m                    |
| Maximum detection range  | 12 km                   |
| Azimuthal angle range    | 0—360°                  |
| Elevation angle range    | -10—190 °               |

A Doppler lidar can measure radial wind speed along the beam based on the Doppler effect.
From radial wind speed and direction, we can retrieve profiles of wind velocity, wind direction,
and turbulence intensity, as explained in detail in Sect. 2.3.
### 2.2  Scanning settings
The lidar systems can be programmed to scan the surrounding atmosphere. The scan strategy
used in this work is described as follows:




- Every 15 minutes there were 360° revolution velocity-azimuth-display (VAD) scans at elevation angles of 15° and 75°
- Between VAD scans, vertical stares, at 90° elevation angle, were performed continually
- Every day, early in the morning, a hard target scan was performed at a low elevation angle checking pointing accuracy of the lidar

The lidar measures the radial velocity, or Doppler velocity, along the line of sight (LOS), and both vertical stare and VAD scans can acquire information on turbulence intensity. In the first step, we will focus on VAD scans and an approach which has been used extensively in the literature and operation (Frehlich et al., 2006; Hon and Chan, 2014; Thobois et al., 2015). The horizontal wind components, which is of importance for aviation safety, can be retrieved from VAD scans. Data from vertical stares will be used as verification reference in this study.

## 2.3 Theory on Turbulence estimation

The turbulent eddy dissipation rate (EDR, ε) can be used as a turbulence intensity indicator as it represents the conversion of turbulent kinetic energy to heat(Cohn, 1994; Hocking, 1985). There are several approaches to retrieve EDR value from lidar observations. (Frehlich and Cornman, 2002) estimated EDR and the length scale from velocity data. (Oude Nijhuis et al., 2018) compared different methods to retrieve EDR from wind velocity, from Doppler radar though. (Thobois et al., 2015) explored the possibility to estimate EDR by using a Leosphere Windcube lidar in Toulouse, France. In this study, we developed an algorithm to estimate EDR by using the velocity structure function, and applied the algorithm on our lidar in Reykjavik. This method is based on the Kolmogorov theory (Frehlich, 2001), assuming the atmosphere is isotropic and homogeneous over the observation domain.

The radial velocity $V_r$, as measured by the Doppler lidar, can be given by:

$$V_r = U sin\varphi cos\theta + V cos\varphi cos\theta + W sin\theta \quad \text{(Eq. 2.1)}$$

Where $U$, $V$ and $W$ are the wind components in x, y and z direction, $\varphi$ is the azimuthal angle, where 0° points to the north, and $\theta$ the elevation angle, where 90° points vertically (Fig. 1).


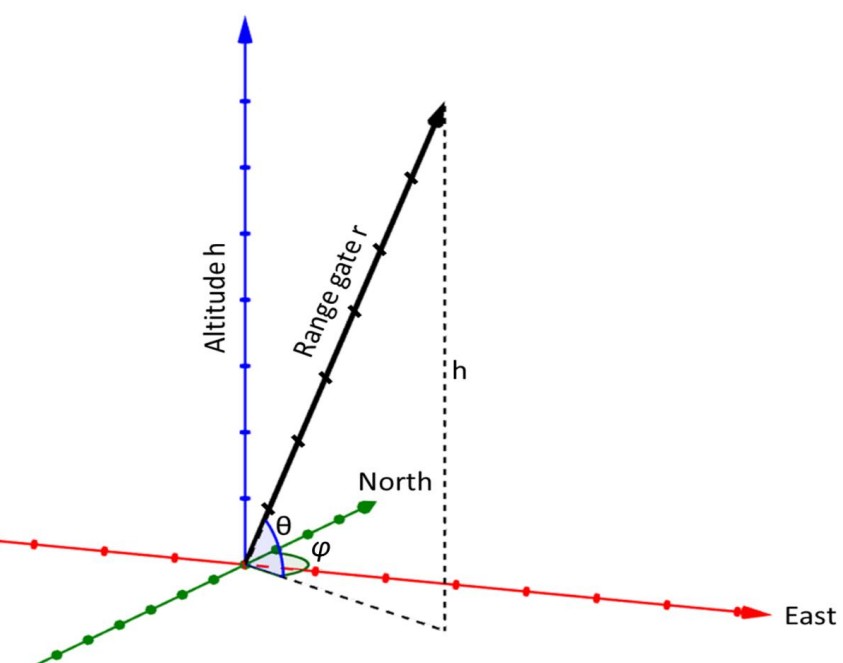

*Figure 1. A schematic of the lidar beam vector (black). The azimuth angle, φ, is shown in*
*green with north being 0° and the elevation angle, θ, in blue with 90° representing a vertical*
*beam. h is the altitude of the range gate, and r is the measurement distance along the beam.*
If the atmosphere is homogeneous and isotropic, the Kolmogorov theory can be applied and
the energy spectrum will fit the -5/3 slope (Fig. 2) as follows:
$$E(k) = C\varepsilon^{2/3}k^{-5/3} \quad \text{(Eq. 2.2)}$$
where $C$ is Kolmogorov constant, $\varepsilon$ is the energy dissipation rate, $k$ is the wavenumber.

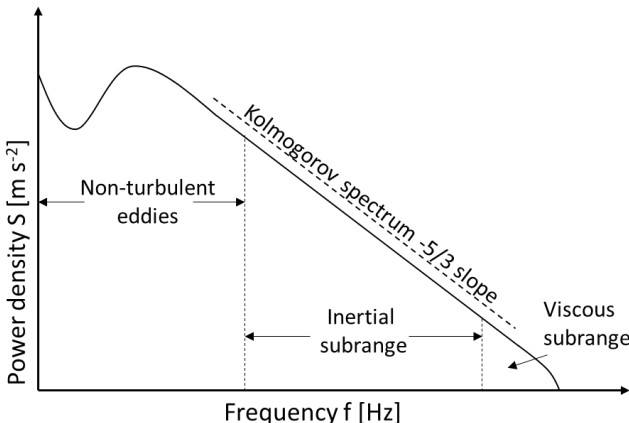





*Figure 2. A schematic of the wind velocity power density as a function of frequency*
*conforming to Kolmogorov's hypothesis (redrawn from (O'Connor et al., 2010; Thobois et al.,*
*2015)).*
If the atmosphere is isotropic and the liner relationship between the power density and the
frequency is close to -5/3 (Figure 2), a direct relationship between the energy spectrum $E(K)$
and the structure function $D_v$ can be defined (Frehlich et al., 2006; Thobois et al., 2015). For a
scanning lidar, the EDR ($\varepsilon$, in unit $m^2 s^{-3}$) can be obtained by fitting the -5/3 slope for the
structure function; then we have
$$D_v = C_v \varepsilon^{2/3} s^{2/3} \qquad \text{(Eq. 2.3)}$$
where $C_v \sim 2$  is the Kolmogorov constant.
The velocity structure function $D_v$ is given by:
$$D_v = < (v'(r) - v'(r+s))^2 > \qquad \text{(Eq. 2.4)}$$
where
$$v'(r) = v(r) - < v(r) > \qquad \text{(Eq. 2.5)}$$
are the fluctuations from the mean velocity $< v(r) >$ at a certain range gate r (Frehlich et al.,
2006) and $s$ is the spatial difference. The distance to range gate r can easily be converted to
height h by using trigonometric functions:
$$h = r sin\theta \qquad \text{(Eq. 2.6)}$$
The structure function can be estimated either along a transverse direction (one azimuth angle
to another), which is the azimuthal approach, or along the lidar beam direction (one range gate
to another), which is the longitudinal approach. Accordingly, $s$ will vary based on this
approach. The comparison between the two approaches will be presented in Sect. 3.2.
Besides the VAD scans, data from the vertical stares were used in this study. (O'Connor et al.,
2010) developed a method to estimate EDR, based on the velocity variance from vertical
stares. This method will be applied in our case and is used as validation reference.


# 3 Results
## *3.1 Noise filtering*

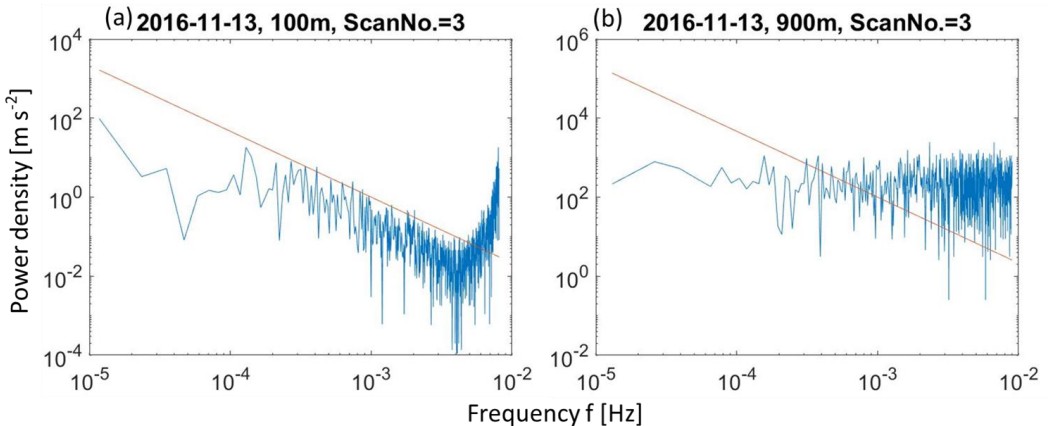

*Figure 3. Verification of data quality: the velocity energy spectrum (blue curve, derived from*
*vertical velocity profile) should correspond to a -5/3 slope (red line). A good case (a) versus a*
*bad case (b) from the same day (13 November 2016) data with different altitude (100 m for*
*(a) and 900 m for (b)).*
First, we investigate observed velocity energy spectra, derived from vertical stare data using
Fast Fourier Transform (FFT) method, to confirm that it agrees with the expected -5/3 slope.
The results vary from one profile to another, but in general, the energy spectrum fits the
idealized slope better at lower altitude. Figure 3 shows two examples of one hour vertical
stare data on 13 November 2016, at different altitudes (100 m and 900 m). The bias between
the signal (blue) and the theoretical slope (red) is likely a result of signal noise at different
heights. The lidar signal is backscattered by fine particles in the atmosphere, thus signal
quality is better at lower altitude, where there are more particles, than higher altitude
(Ramanathan et al., 2001). In high latitude regions like Iceland, the mixing layer is shallower
than at intensively studied continental mid-latitude sites, and the number of scatters is
relatively smaller. This could amplify this bias as well.
(Manninen et al., 2016) estimated the uncertainty introduced by noise when they quantified
turbulence intensity via lidar data and they developed a background correction algorithm to
increase data availability. We have applied this algorithm on the lidar data from vertical stare,
but it is not implemented for VAD scans yet. To distinguish valid data from unrealistic data
points, or noise, we applied a filter based on the carrier-to-noise ratio (CNR) and lidar
confidence index (CI). CNR indicates the quality of the data received. Higher CNR means a
better received signal. Confidence index (0 to 100) is a built-in variable provided by the lidar
manufacturer for quality control. There are two ways to apply the filters: before EDR
calculation or after EDR calculation. The different filtering methods are named as shown in
Table 2.
*Table 2. Filter methods tested in this study.*



|  | Before-calculation filter | After-calculation filter |
|---|---|---|
| CI only | CI-B | CI-A |
| CNR only | CNR-B | CNR-A |
| CNR and CI | Combi-B | Combi-A |

Applying the filter before calculation may remove most of the noise, but also some realistic
observations, while after calculation the filter may keep some unexpected noise, especially
with the longitudinal approach. Figure 4 shows the effect of different filters. Figure 4(a) and
4(b) show CNR and CI on 24 March 2017. The calculated EDR (longitudinal approach)
without any filter is shown in Fig. 4(c). Figure 4(d), 4(e), and 4(f) are filtered results of 4(c),
with Combi-A, CI-B, and Combi-B filters, respectively. The thresholds for CNR and
confidence index filter are -32 dB and 99 in this case. The CNR profile of the lidar data can
vary greatly, which makes it risky to have a constant threshold: a high threshold may remove
too many data points while a low threshold may keep too much noise. The optimal CNR
threshold was, based on numerous tests, identified to be -32 dB at which the data availability
is maximized and at the same time the noise minimized. Before-calculation filters are more
sensitive to this threshold, compared to after-calculation filters.

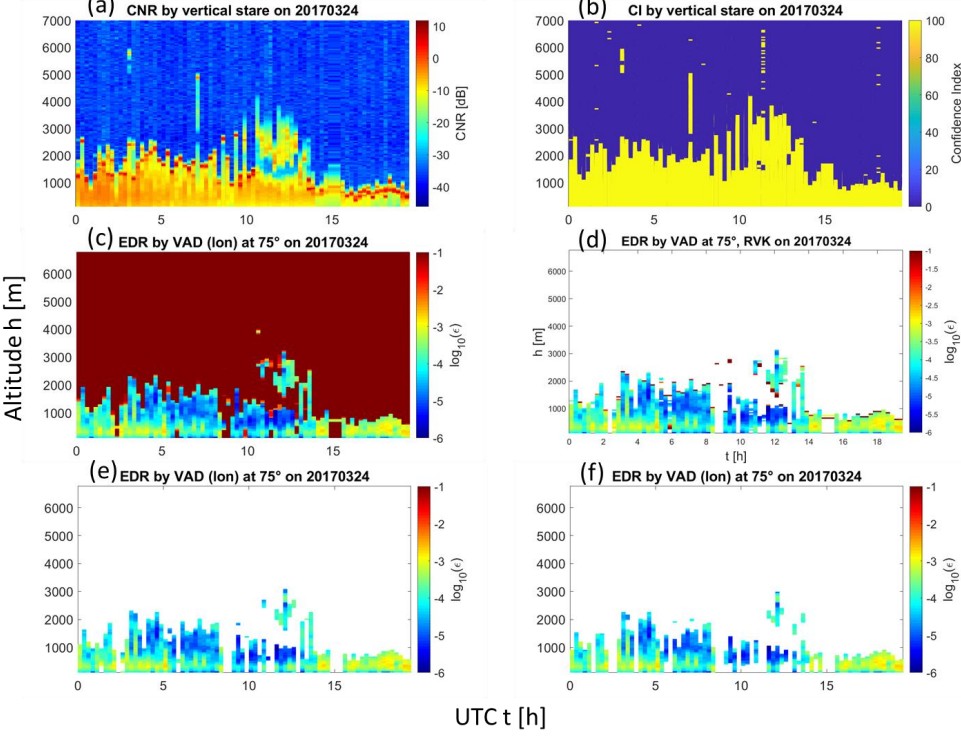

*Figure 4. The carrier-to-noise ratio (CNR) in dB (a), the Confidence Index(CI) (b), the base*
*10 logarithm of EDR [m² s⁻³] without filter (c), with Combi-A filter (d), CI-B filter (e) and*



*Combi-B filter (f) on 24 March 2017. The EDR is derived from VAD scans, with 75° elevation*
*angle and using longitudinal approach.*
The impact of different filters is stronger for single VAD scans. Figure 5 compares EDR
along each beam by filtering after calculation (Combi-A, left panel) and filtering before
calculation (Combi-B, right panel) and gives three examples at 00:14 (5(a) and 5(b)), 12:29
(5(c) and 5(d)) and 01:14 (5(e) and 5(f)). In the last case there is no data with the before-
calculation filter. In most of the scans, noise at the farthest ranges is removed when filtering
before calculation, as seen by comparing Fig. 5(a) and 5(b). However, there can be some
over-filtering with before-calculation filter, as seen by comparing Fig. 5(c) and 5(d). Some
valid data points have been removed alongside with noise and only data of several range bins
remains. In some extreme cases with clear sky conditions (e.g. Fig. 5(e) and 5(f)), the before-
calculation filter will remove all data. This is the case when aerosol concentrations are too
low to provide enough backscatter data. For an experienced researcher, it is not hard to
distinguish the noise, but that may not be the case for untrained users. Thus we would suggest
to apply the before-calculation filter for operational purpose because of its higher reliability,
but the after-calculation filter for research purpose, since more usable data points will be kept.
In the rest of this paper we use the filter after calculation (Combi-A) if not specified
otherwise.

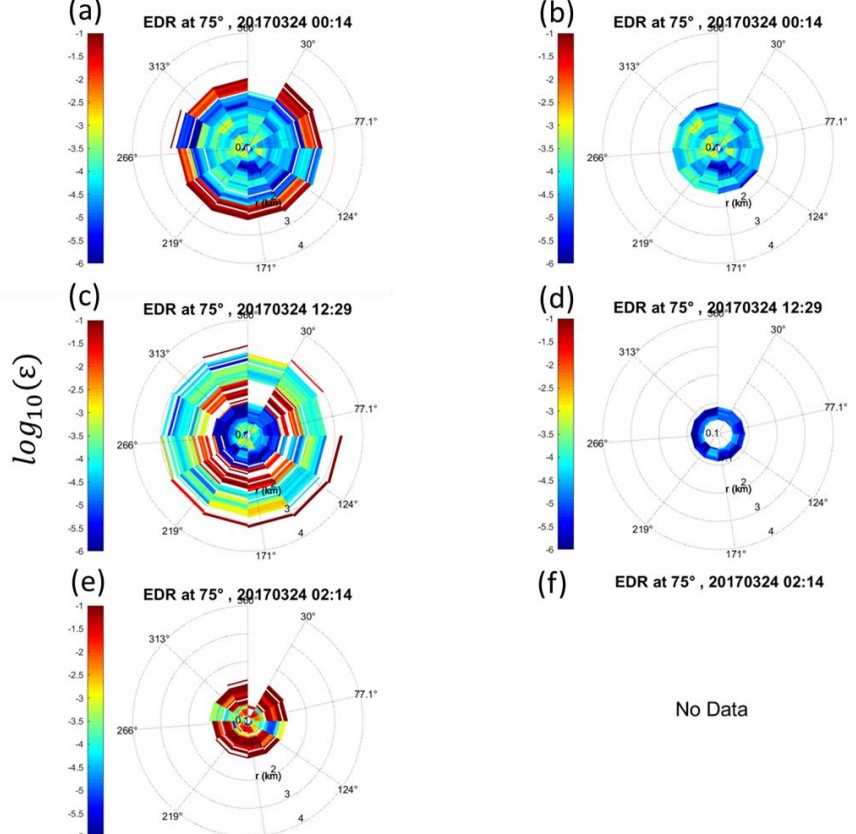




*Figure 5. Base 10 logarithm of EDR [m² s⁻³] retrieved by longitudinal approach at three*
*times (top: 00:14, centre: 12:29 and bottom: 02:14) on 24 March 2017, with the after-*
*calculation filter (a, c, e) and before-calculation filter (b, d, f). There is no remaining data at*
*01:14 if we use the before-calculation filter.*

## 3.2 EDR estimation

We have applied the EDR retrieval algorithm to data from the lidar at IMO. We choose two
examples in this study: 24 March 2017 as a turbulent case and 31 March 2017 as a calm case.
Figure 6 compares the derived EDR using different scan strategies (vertical to VAD),
structure functions (azimuthal to longitudinal approach), elevation angles (75° to 15°), and in
different weather conditions (turbulent and calm). The received lidar backscattered signal is
directly related to the size and amount of particles in the atmosphere. Higher concentration or
larger particles means higher received signal, or higher CNR values (see Fig. 4). From the
Fig. 4 and previous research (O'Connor et al., 2010), we know most of the signal come from
the well mixed boundary layer which height can vary from tens of meters to a few kilometers
(Stull, 1988). Above the boundary layer, the CNR is usually lower and the signal is dominated
by noise. In general, one would expect the boundary layer to be thicker when the conditions
are turbulent than in calm conditions, due to stronger vertical mixing process. This explains
why there is a clear difference in detection height between the more turbulent day (Fig. 6, left
panel) when the detection height is about 3.5 km altitude and the calm day (Fig. 6, right
panel) where it is only about 1.5 km. Normally, the lidar can capture the efficient backscatter
signal up to the top of boundary layer, but sometimes we can also see the strong signal above
it, like the ice cloud around 12 UTC on 24 March.





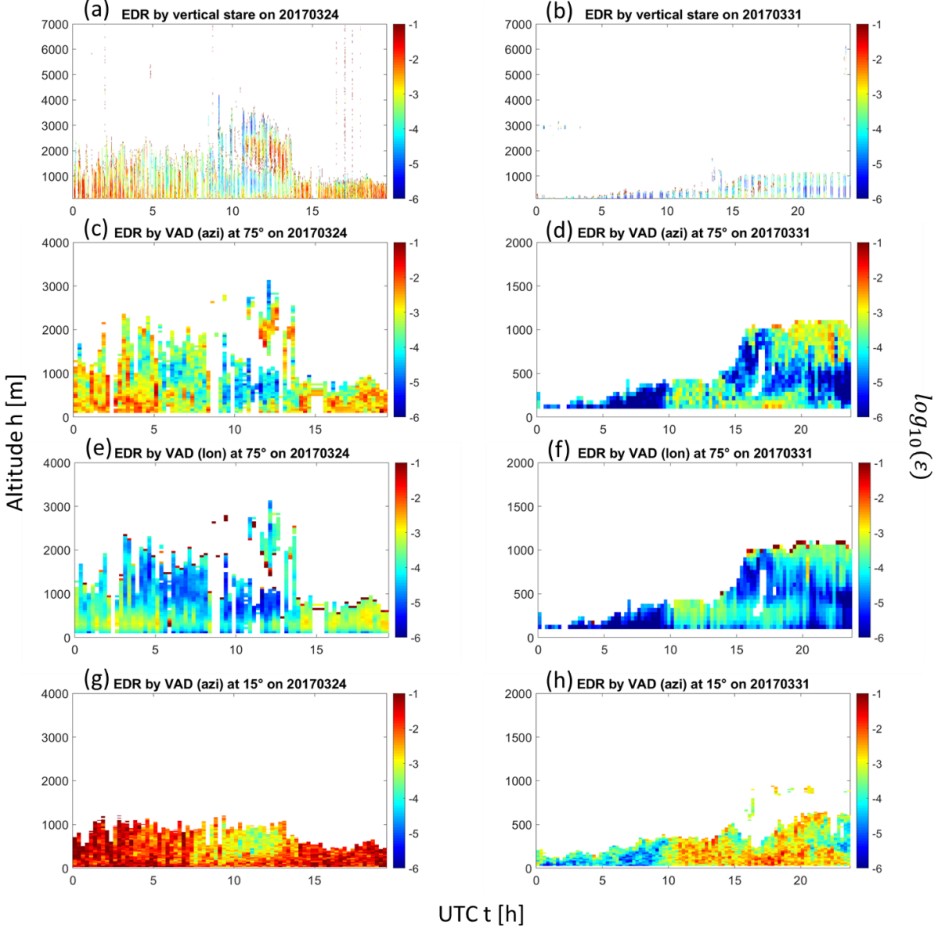

Figure 6. Comparison of base 10 logarithms of eddy dissipation rate (EDR [$m^2 s^{-3}$]) on 24
March 2017 (left panels) and 31 March 2017 (right panels). EDR is derived from the vertical
stare (a and b), VAD scans using the azimuthal approach (azi) with 75° elevation angle (c
and d), longitudinal approach (lon) with 75° angle (e and f), and azimuthal approach with
15° elevation angle (g and h). Note the different vertical axis.

As seen in Fig. 6, the EDR retrievals using the structure function on VAD scans are
quantitatively similar to the EDR retrievals from the vertical stare applying the method of
(O'Connor et al., 2010). In the early morning and late night of 24 March, there are high
values of EDR in four figures (Fig. 6, left panel), indicating turbulent conditions. On the calm
day, 31 March, turbulence is detected after 10 UTC in all four figures (Fig. 6, right panel),
and we can also see that in the afternoon, from 15 UTC, the top and bottom of the boundary
layer was more turbulent than the middle of the boundary layer. In Fig. 6(e), an EDR
maximum layer can be identified around 250 m, and similar results can be found in some
other days' longitudinal data (not shown). This does not mean that there is a constant layer at
that altitude, because it may be found from the data with 15° elevation angle, but at lower
altitude. This cannot be found in azimuthal approach, so we tend to exclude the reason from

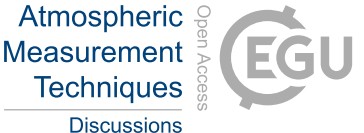

lidar itself. The likely reason is the better data availability at lower altitude which may result
in the moving average algorithm introducing this pseudo-layer in some certain weather
situation, which is still unknown to us. The comparison between azimuthal (Fig. 6(c) and
6(d)) and longitudinal (Fig. 6(e) and 6(f)) approaches shows that they yield similar turbulence
patterns, both in the turbulent and in the calm case. However, the longitudinal approach,
besides the unfiltered noise signal at the edge of different columns, which we have discussed
in Sect. 3.1, yields lower absolute EDR values. In other words, some severe turbulent events
have been underestimated. This is more obvious in the turbulent case (24 March). This results
from our time series analysis: we apply one more averaging calculation on the longitudinal
approach than on the azimuthal approach, averaging the EDR at each range gate, to plot the
altitude to time figure. In this way, some turbulence will be averaged out, or smoothed. As
turbulence is a sudden change of airflow on a small temporal and spatial scale, more
averaging calculations means less turbulence is detected. In this regard, the azimuthal
approach performs better on EDR retrieval. As mentioned above, we use the VAD scans at
two elevation angles, 15° and 75°. Due to the basic geometry, higher elevation angle means
higher upper and lower detection limits. Thus, a low elevation scan could provide more
information at lower altitude. The scans at different elevation angles do not execute
simultaneously, and we do not expect a perfect match of EDR. However, the results of the 15°
elevation angle (Fig. 6(g) and (h)) shows significant higher EDR value than at 75°. One
possible explanation is that the lower elevation angle results in higher vertical resolution (25.9
m to 96.6 m at 75°), which means better ability to distinguish small scale turbulence.
However, the heterogeneity of the atmosphere should also be kept in mind. With a low
elevation angle, further range gates also mean a larger distance from the zenith position,
resulting in lower representativeness and higher uncertainty. For example, at 75° elevation
angle, the straight line distance between two adjacent data points at 1000 m range gate would
be 517 m, while at 15° elevation angle it is 1931m. Besides, it is also possible that more
turbulence happens near surface due to friction. We recommend combining both scan angles
operationally: the low elevation scan gives high resolution measurement at lower altitude, and
the high elevation scan provides information with lower uncertainty at higher altitude.
Although the azimuthal approach performs better on the time series analysis, the longitudinal
approach has its own advantage: it can retrieve EDR values along the beam, which allows us
to examine where the turbulence occurs on a horizontal scale. Currently, we have two VAD
scans at different elevation angles every 15 minutes. This enables us to detect in which
direction and at which distance from the lidar turbulence is occurring. Figure 7 shows two
examples from 24 March 2017: one is at 08:00 UTC when the atmosphere was relatively
calm, as seen in Fig. 6, and another is at 16:14 UTC when the conditions were more turbulent.
Ignoring the hot-spots at the edges, which are noise as we have discussed, we can see where,
in horizontal space, turbulence is detected. In the morning, the turbulence mostly occurred at
a lower level and close to the lidar. This can be seen at  the 15° angle but not 75°, because the
first range gate is at 100 m, corresponding to 96.6 m altitude at 75° but only 25.9 m altitude at
15° angle. In this case, the turbulence is occurring below 96.6 m and the VAD scan at 75°
cannot detect it. In the afternoon, more turbulence is detected (higher EDR values), and this
agrees well with Fig. 6.



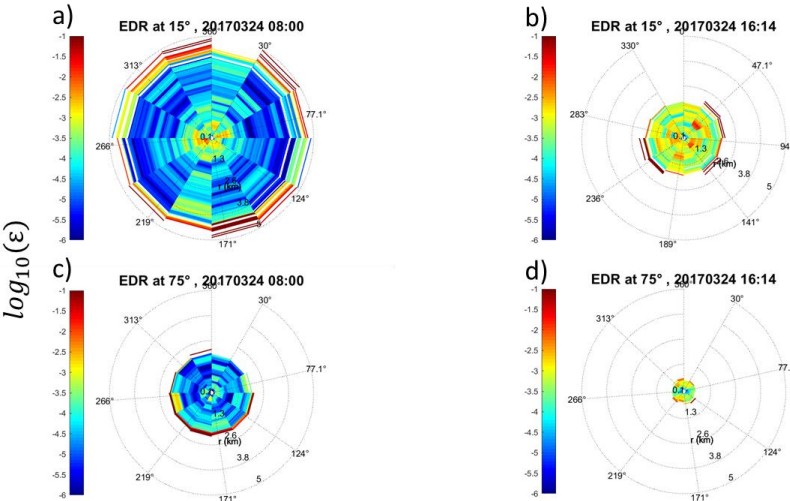

*Figure 7. Beam-circular EDR [m² s⁻³] maps for two cases at 08:00 UTC (left) and 16:14 UTC*
*(right) 24 March 2017. The data is from the VAD scan, at elevation angle 75° (top) and 15°*
*(bottom), using the longitudinal approach. The radial direction indicates the distance from*
*the lidar.*

The azimuthal interval of current scanning is 30°, thus the distance between two data point
could be larger than the scale of turbulence, which means the assumption of homogeneous
atmosphere is less valid as we discussed above and the turbulence with small scale might be
ignored. Potentially the azimuthal interval could be decreased but further research is needed
on the effect of that on the EDR and the associated uncertainties.  Further future
considerations would also be to set up scans along airport runways. This has been done at
some airports, such as Hong Kong airport (Hon and Chan, 2014). The results from our study
will be valuable to air traffic controllers in Iceland, where weather and atmosphere conditions
are distinctly different from Hong Kong. We have applied the algorithm to the other lidar
system at Keflavik International airport, and the results are also online and accessible to the
air traffic controllers in Iceland.

## 4  Conclusion

The current work suggests that it is possible to apply VAD scans and a structure function to
derive an estimate of the EDR. The results are comparable to more conventional EDR
retrievals using vertical stares. The algorithm works in different weather conditions. Iceland is
in the subpolar region, and the data quality is not as good as it is in some other regions due to
a lack of scatters, thus data quality control is especially important. Aerosol concentration in
the atmosphere at high latitudes is often lower than at mid and low latitudes, which results in
a weaker backscattered signal and the noise may be misinterpreted as severe turbulence.
Finding a reliable algorithm to filter noise is thus important. Simply filtering data by the
carrier-to-noise ratio is not reliable, since the signal strength can vary considerably from day
to day, profile to profile. Besides, CNR values change with range, which makes it difficult to
find a universal threshold. Applying a filter before or after the EDR calculation also makes a
difference: with the same threshold, before-calculation filter remove most of the noise and
some valid data, whereas after-calculation filter does not remove all of the noise. It is



important to find a balance where as much of the noise as possible is removed while simultaneously as many valid data points are kept. In the current phase, we would suggest using the after-calculation filter for research purpose as it will keep more data points, and using the before-calculation filter for operational purposes as it removes most of the noise. We also tested two approaches to calculate the structure function from the VAD scan, and they perform differently: the azimuthal approach performs better (in comparison to the vertical stare) in time series analysis, and indicates when and at what altitude the turbulence occurs, while the longitudinal approach is more suitable to show location of the turbulence relative to the lidar on a horizontal scale. The EDR values vary with different VAD elevation angle, thus a combination of both angles is recommended. Uncertainties are also introduced in a more theoretical way. To apply the Kolmogorov theory, we have to assume the atmosphere is homogeneous and isotropic, but as we have described, the VAD scan is cone-shaped, which means the assumption is only valid at lower range gates and at larger elevation angles. To apply this method in an operational way, we would suggest combining the VAD scan with two elevation angles: the lower angle scan can fill the gap between the surface and the first range gate of the higher angle scan, while the higher angle scan provides reliable information at higher altitude.

In general, the method of retrieving EDR from lidar data to estimate the turbulence intensity is possible and it can be applied to the air traffic control system in Iceland. Furthermore, at present there is ongoing work to make this algorithm operational and to find a solid threshold of different EDR value categories in practical use.

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
