# Peer review of "Using Doppler lidar systems to detect atmospheric turbulence in Iceland"

_Atmospheric Measurement Techniques, 2019_

## Referee Comment (RC1) · Anonymous Referee #1 · 22 Feb 2019

The paper presents the new lidar measurements of eddy dissipation rate at Reykjavik, iceland, using commercially available scanning Doppler lidar. This atmospheric dynamic parameter is highly important for both dynamic studies and aviation safety, as the author points out. However, I am concerned about the presented EDR results, since this work lacks the data quality discussion. The two highly important parameters, CRN and CI, are poorly defined, and there is not nearly enough discussion on the measurement uncertainty of the so called structure function Dv. The statistical uncertainty, due to inadequate sampling is also not mentioned at all in the paper. The decision of filtering after or before calculation also seems arbitrary without scientific algorithm to justify. Thus, unless the author can provide new information on these uncertainty issue, these results and conclusions in this paper is unacceptable for the journal publication.

[Figure]

Technique issues: 1. ) The titles for most of the plots seem to be added manually afterwards and become out of place. Most of them blocked the top part of the plots. 2.) Why isn't there tick marks and unit in x-axis of Figure 2? 3.) How do you explain the reversal of power density at high frequency in Figure 3? Is it data quality related or does it indicate some atmospheric dynamic feature? 4.) On page 7, line 13, the author talks about different data quality at low altitude and high altitude. It would be helpful to show some statistics of good data at low altitude (such as at 100 m). 5.) Why does the author show all these high altitude results with low CNR in these contour plots, such as Figure 4 and Figure 6. I would choose a consistent altitude range (1-3000 m, maybe) throughout the paper. 6.) On page 10, line 17-18, "In general...", this sentence does make sense to me. 7.) on page 12, line 8, should be "These results".
* * *

---

## Author Comment (AC1) · 5 Mar 2019

Dear Anonymous Referee #1,

Thank you very much for your careful review and constructive comments of our manuscript. We really appreciate the suggestions you gave, which will help us to improve this manuscript. We are currently working on revising the manuscript according to your comments, which may take some time. Here we would like to give a short reply to your comments. Please check the supplement via the link below.

With kind regards,

Authors

Please also note the supplement to this comment:
https://www.atmos-meas-tech-discuss.net/amt-2019-3/amt-2019-3-AC1-supplement.pdf

**Supplement:**

Dear Anonymous Referee #1,

Thank you very much for your careful review and constructive comments of our manuscript. We really appreciate the suggestions you gave, which will help us to improve this manuscript. We are currently working on revising the manuscript according to your comments, which may take some time. Here we would like to give a short reply to your comments.

With kind regards,

Authors

General comments:

1. *The two highly important parameters, CNR and CI, are poorly defined …*
   Response: We agree with the reviewer and will provide a thorough definition of both parameters, CNR and CI, in a revised version of the manuscript.

2. *… there is not nearly enough discussion on the measurement uncertainty of the so called structure function Dv. The statistical uncertainty, due to inadequate sampling is also not mentioned at all in the paper*
   Response: We agree and will provide an extensive discussion on the uncertainty of EDR by performing an uncertainty analysis in the revised version of the manuscript. We are planning to use the analytic error propagation approach to investigate the uncertainty.

Technique issues:

1. *The titles for most of the plots seem to be added manually afterwards and become out of place. Most of them blocked the top part of the plots.*
   Response: We will improve or redraw all plots accordingly, making sure that all title, fonts, legend are appropriate.

2. *Why isn't there tick marks and unit in x-axis of Figure 2?*
   Response: Figure 2 was thought more a schematic figure why we didn't use tick marks, but we will take this comment into account in a revised version.

3. *How do you explain the reversal of power density at high frequency in Figure 3? Is it data quality related or does it indicate some atmospheric dynamic feature?*
   Response: We will investigate the power density at high frequency during other days and analyze possible reasons for the reversal. A discussion on this topic will be included in the revised version.

4. *On page 7, line 13, the author talks about different data quality at low altitude and high altitude. It would be helpful to show some statistics of good data at low altitude (such as at 100 m)*
   Response: As the referee suggested, this will be done in the next version.

5. *Why does the author show all these high altitude results with low CNR in these contour plots, such as Figure 4 and Figure 6. I would choose a consistent altitude range (1-3000 m, maybe) throughout the paper.*
Response: In Fig. 4, our initial idea was to show the data filtering results, so we simply presented the full range. We agree with the reviewer that having a consistent altitude range improve the quality of the figures and the boundary layer height will be clearly visible. We will redraw all plots using 3 km as the maximum altitude.

6. *On page 10, line 17-18, "In general...", this sentence does make sense to me.*
Response: This sentence will be rephrased as: *As we expected, the boundary layer is thicker when the conditions are turbulent than it in calm conditions, due to a stronger vertical mixing process.*

7. *on page 12, line 8, should be "These results".*
Response: This will be corrected.

---

## Referee Comment (RC2) · Anonymous Referee #2 · 10 Jun 2019

This manuscript implements different methods for deriving turbulent parameters from scanning Doppler lidar observations in the lower atmosphere. Retrieving turbulent parameters is of high interest, both scientifically and operationally, as described in the manuscript introduction. Data from this challenging location is also of major interest.

The topic is introduced well but the promise of the paper is not fulfilled in the remaining sections. The methodology and results sections are quite sparse and the discussion on error sources and method intercomparisons, in particular, inadequate. The plots also need to be improved.

Please include more information on the instrument and scanning parameters employed (e.g. for VAD scans, how many azimuths per scan and how many pulses integrated per azimuth; how many pulses integrated per profile for vertical stare). These are

important for estimating the uncertainties associated with the measurements, and then the retrieved quantities. Uncertainty in the velocity estimate depends on the carrier-to-noise ratio, CNR, therefore, a single CNR threshold is valid for determining reliable velocity estimates. The theoretical relationship between CNR and velocity uncertainty given by the Cramer-Rao lower-bound method can be calculated from the instrument parameters (e.g. Rye and Hardesty, 1993), and this should be calculated and plotted for the instrument used in this study as it provides the basis for the uncertainties, filters and thresholds used. The definition of CNR and CI presented in the paper should be improved.

There are plenty of papers discussing different techniques for retrieving turbulent parameters from Doppler lidar, and the expected uncertainties (e.g. Sathe and Mann, 2013; Banakh et al., 2017; Smalikho and Banakh, 2017). These also show how the uncertainties from the initial velocity estimates propagate through to the retrieved tubulent quantities. This would then aid the intercomparison of the different methods.

Some additional issues points to consider: Figure 2 and text. It should be stated that the energy spectrum will follow the expression given in Eq 2.2 for frequencies within the inertial subrange. There are no values on the axes - add these or give typical values for the different ranges.

Figure 3. Please provide the typical CNR values for the 'good' and 'bad' cases. I am surprised by the increase in power density for high frequencies in Fig. 3a; is this noise or an atmospheric feature? I would usually expect the noise contribution to be 'white noise' which is spectrally flat, as is seen in Fig. 3b. Time-height plots of the original signal and Doppler velocity values that these spectra are taken from would aid interpretation.

Please describe the tests you made to determine the optimal CNR threshold of -32 dB. What does this mean in terms of velocity and dissipation rate of turbulent kinetic energy (EDR) uncertainty estimates? It should be noted that, since EDR depends on

[Figure]

the velocity variance, the reliability threshold for good EDR estimates may not require a constant CNR threshold, but be described in terms of the relative contribution of velocity uncertainty to the EDR estimate; higher values of EDR can cope with more uncertain velocity estimates.

Figure 5. Without the original signal and velocity plots it is not possible to see the impact of filtering. The range axis is not obvious in these plots; it might be clearer to use standard x-y axes.

Figure 6. It is difficult to compare the methods as the panels don't use the same vertical axes. It is interesting to see that the different methods show similar features but, in most cases, very different values.

Figure 7. If it is known that the 'hot spots' are noise, why are these not filtered? Why do the filters applied not capture these? Or should additional filters be applied? Figure 7 could be plotted with a maximum range of 2 km, then the features in the regions of good signal would be more evident.
* * *

---

## Author Comment (AC2) · 7 Jul 2019

Dear editor, dear referees,

First of all, I would like to thank the editor and the two anonymous reviewers for your careful review and constructive comments of our manuscript. We really appreciate the detailed suggestions you gave, which definitely helped us to improve this manuscript. Personally, this is my first manuscript and all your reviews where instrumental in improving the manuscript. As the author group, we have tried to compile the comments as fully as possible. We have revised the manuscript and some of the major changes are:

- The introduction of a new variable $R^2$ as an indicator of atmospheric homogeneity (Sect. 2.3).
- The removal of the comparison of different filtering methods. Old Fig. 5 is removed, which compared before- and after- calculation filtering method.
- The subsection *Sect. 3.1* has been altered from *filtering noise* to *data screening*. Now, a sensitivity test is presented (new Fig. 4) on different CNR threshold. A comparison is made between CNR and CI thresholds.
- The addition of a new subsection about error analysis (Sect. 3.2), using the method from Smalikho and Banakh (2017).

More detailed information can be found in the revised manuscript. The responses to each comment are found below.

With kind regards,

Yang Shu

On behalf of all authors

To Anonymous Referee #1

General comments:

1. *The two highly important parameters, CNR and CI, are poorly defined …*
   Response: We have now included a detailed definition and explanation of CNR and CI in Sect. 3.1. Furthermore, we present a sensitivity test on different CNR thresholds (Fig. 4) and compare the impact of CI and CNR (Table 2). More details can be found in the revised manuscript. The text about the definition is shown below
   Change in Manuscript:
   *…we use Carrier to Noise Ratio (CNR) as one indicator of the backscatter signal intensity. The CNR value depends on the concentration of aerosols in the atmosphere that backscatter laser light. High atmospheric backscatter coefficient leads to high CNR. Weather conditions also impact the CNR level. In the cases shown in Fig. 3, the mean CNR is -4.3 dB at 450 m and -32 dB at 1950 m. Low CNR signal means the signal is dominant by noise instead of real information, thus CNR is used for data screening in many studies (Gryning et al., 2017a, 2017b). Besides CNR, a confidence index (CI) is applied, which is one of the outputs of the Leosphere Windcube, alongside radial wind and CNR data, to screen noise and invalid data points. Radial wind at each time- and range-step is determined by computing the spectrum using an FFT method, and subsequently fitting this spectrum to a theoretical curve. The CI threshold depends on CNR, mean error and spectrum broadening of this spectral fit. CI is factory calibrated individually for each*

*lidar system and each range gate length. The calibration requires a few hours of noise measurements, where outgoing radiation is shielded from the receiver telescope. The CI threshold is then set to a value that limits the false positive rate to 0.25% (Dabas, 1999). For the scans applied here CI is a binary quality control parameter returning the value 0 for rejected data points and 100 for valid data points*

2. *… there is not nearly enough discussion on the measurement uncertainty of the so called structure function Dv. The statistical uncertainty, due to inadequate sampling is also not mentioned at all in the paper*
   Response: The reviewer is absolutely correct. We have included relative error as an indicator of uncertainty analysis, using the method from Smalikho and Banakh, 2017. The measured velocity uncertainty is related to atmospheric homogeneity: if the atmosphere is homogeneous, the sine curve fits better, and the results are more reliable. Here we compared $R^2$ value for different scans and CNR thresholds in Sect 3.1. The main result of error analysis is shown below.
   Change in Manuscript: *… The relative error of EDR reaches 20.2% of the calculated EDR for 75° elevation angle and 9.8% for 15° elevation angle. These results are comparable to the results from Smalikho and Banakh (2017), which are 15~20%. …*

Technique issues:

1. *The titles for most of the plots seem to be added manually afterwards and become out of place. Most of them blocked the top part of the plots.*
   Response: All figures have been replotted. The font size has been changed. The titles, legends, etc. were generated together with the figures for consistency.

2. *Why isn't there tick marks and unit in x-axis of Figure 2?*
   Response: The unit has been added. As the inertial subrange is theoretically defined for three-dimensional turbulence as the range where the spectrum is proportional to $k^{-5/3}$ there are no typical values, but the values depend on local situation and measurement. Fig. 3 shows an example from our data with the inertial subrange starting at around 102 Hz. The annotation of Fig. 2 has been modified as follow.
   Change in Manuscript: *Figure 2. A schematic of the wind velocity power density as a function of frequency conforming to Kolmogorov's hypothesis, redrawn from O'Connor et al. (2010) and Thobois et al. (2015). The inertial subrange is the part of the power spectrum where energy is transferred to smaller scales by turbulence. For three-dimensional turbulence, the power spectrum is theoretically proportional to k-5/3, where k is the wave number.*

3. *How do you explain the reversal of power density at high frequency in Figure 3? Is it data quality related or does it indicate some atmospheric dynamic feature?*
   Response: The increase at high frequencies has been checked, and it seems that there was an issue with the power spectra calculation, which has been corrected and updated in the revised version as well. Now we also use the data on 24 March 2017, same with other analysis (in the old version we use data from 2016).

4. *On page 7, line 13, the author talks about different data quality at low altitude and high altitude. It would be helpful to show some statistics of good data at low altitude (such as at 100 m)*
   Response: The data quality and availability difference between high and low altitude are compared in Fig. 4 now.

5. *Why does the author show all these high altitude results with low CNR in these contour plots, such as Figure 4 and Figure 6. I would choose a consistent altitude range (1-3000 m, maybe) throughout the paper.*
   Response: We agree that a consistent altitude range would be better. Now we use 3 km as the maximum altitude for the Fig. 5,6,7.

6. *On page 10, line 17-18, "In general...", this sentence does make sense to me.*
   Response: This sentence has been rephrased.
   Change in Manuscript: *As expected, the boundary layer is thicker in turbulent conditions compared to calm conditions, due to a stronger vertical mixing process.*

7. *on page 12, line 8, should be "These results".*
   Response: Corrected.

To Anonymous Referee #2

1. *Please include more information on the instrument and scanning parameters employed (e.g. for VAD scans, how many azimuths per scan and how many pulses integrated per azimuth; how many pulses integrated per profile for vertical stare). These are important for estimating the uncertainties associated with the measurements, and then the retrieved quantities. Uncertainty in the velocity estimate depends on the carrierto-noise ratio, CNR, therefore, a single CNR threshold is valid for determining reliable velocity estimates. The theoretical relationship between CNR and velocity uncertainty given by the Cramer-Rao lower-bound method can be calculated from the instrument parameters (e.g. Rye and Hardesty, 1993), and this should be calculated and plotted for the instrument used in this study as it provides the basis for the uncertainties, filters and thresholds used. The definition of CNR and CI presented in the paper should be improved.*
   Response: We agree and in accordance with a similar comment from Referee #1 we added more information in Table. 1 and Sect. 2.2. A sensitive test of CNR threshold has been presented and we selected the single threshold -32 dB. The detailed information is provided in Sect. 3.1. The CNR and CI are better defined as follow (see also response to Referee #1).
   Change in Manuscript:
   *…Pulse rate frequency    20 kHz…*
   *… The transverse interval (azimuthal resolution) is 30°, resulting in 12 beams for each VAD scan. The accumulation time for each beam is 5 s.*
   *Between VAD scans, vertical stares, at 90° elevation angle, were performed continually at 1 s accumulation time per profile…*
   *…we use Carrier to Noise Ratio (CNR) as one indicator of the backscatter signal intensity. The CNR value depends on the concentration of aerosols in the atmosphere that backscatter laser*

*light. High atmospheric backscatter coefficient leads to high CNR. Weather conditions also impact the CNR level. In the cases shown in Fig. 3, the mean CNR is -4.3 dB at 450 m and -32 dB at 1950 m. Low CNR signal means the signal is dominant by noise instead of real information, thus CNR is used for data screening in many studies (Gryning et al., 2017a, 2017b). Besides CNR, a confidence index (CI) is applied, which is one of the outputs of the Leosphere Windcube, alongside radial wind and CNR data, to screen noise and invalid data points. Radial wind at each time- and range-step is determined by computing the spectrum using an FFT method, and subsequently fitting this spectrum to a theoretical curve. The CI threshold depends on CNR, mean error and spectrum broadening of this spectral fit. CI is factory calibrated individually for each lidar system and each range gate length. The calibration requires a few hours of noise measurements, where outgoing radiation is shielded from the receiver telescope. The CI threshold is then set to a value that limits the false positive rate to 0.25% (Dabas, 1999). For the scans applied here CI is a binary quality control parameter returning the value 0 for rejected data points and 100 for valid data points*

2. *Figure 2 and text. It should be stated that the energy spectrum will follow the expression given in Eq 2.2 for frequencies within the inertial subrange. There are no values on the axes - add these or give typical values for the different ranges.*
   Response: As the inertial subrange is theoretically defined for three-dimensional turbulence as the range where the spectrum is proportional to $k^{-5/3}$ there are no typical values, but the values depend on local situation and measurement. Fig. 3 shows an example from our data with the inertial subrange starting at around 102 Hz. The annotation of Fig. 2 has been modified as follow.
   Change in Manuscript: *Figure 2. A schematic of the wind velocity power density as a function of frequency conforming to Kolmogorov's hypothesis (redrawn from O'Connor et al. (2010) and Thobois et al. (2015)). The inertial subrange is the part of the power spectrum where energy is transferred to smaller scales by turbulence. For three-dimensional turbulence the power spectrum is theoretically proportional to k-5/3, where k is the wave number.*

3. *Please provide the typical CNR values for the 'good' and 'bad' cases. I am surprised by the increase in power density for high frequencies in Fig. 3a; is this noise or an atmospheric feature? I would usually expect the noise contribution to be 'white noise' which is spectrally flat, as is seen in Fig. 3b. Time-height plots of the original signal and Doppler velocity values that these spectra are taken from would aid interpretation.*
   Response: The CNR value has been added (*-4.3 dB for the "good" case and -32 dB for the "bad" case*). The increase at high frequencies has been checked, and it seems to be an issue of the power spectra calculation, which has been corrected and updated in the revised version as well. Now we also use the data on 24 March 2017, same with other analysis (in the old version we use data from 2016).

4. *Figure 5. Without the original signal and velocity plots it is not possible to see the impact of filtering. The range axis is not obvious in these plots; it might be clearer to use standard x-y axes.*

Response: Now we remove Fig. 5, since we do not compare different filtering methods anymore. The original data and filtered data can be checked in new Fig. 5. Most figures have been replotted to make each element more visible.

5.  *Figure 6. It is difficult to compare the methods as the panels don't use the same vertical axes.*
    Response: Now the maximum altitude has been set to 3 km for Fig. 5,6,7.

6.  *Figure 7. If it is known that the 'hot spots' are noise, why are these not filtered? Why do the filters applied not capture these? Or should additional filters be applied? Figure 7 could be plotted with a maximum range of 2 km, then the features in the regions of good signal would be more evident.*
    Response: Some noise was kept due to the bad performance of the old filtering method (Combi-A). Now we have removed the comparison of different filtering methods, only applied one filter (Combi-B in the previous version), and the hot spots were filtered. The maximum range has been modified. To make it unified throughout the whole manuscript, we set it to 3 km.